# Towards Model Selection using Learning Curve Cross-Validation

**Felix Mohr**                   FELIX.MOHR@UNISABANA.EDU.CO
*Universidad de La Sabana, Colombia*

**Jan N. van Rijn**                J.N.VAN.RIJN@LIACS.LEIDENUNIV.NL
*Leiden University, the Netherlands*

## Abstract

Cross-validation (CV) methods such as leave-one-out cross-validation, k-fold cross-validation, and Monte-Carlo cross-validation estimate the predictive performance of a learner by repeatedly training it on a large portion of the given data and testing on the remaining data. These techniques have two drawbacks. First, they can be unnecessarily slow on large datasets. Second, providing only point estimates, they give almost no insights into the learning process of the validated algorithm. In this paper, we propose a new approach for validation based on learning curves (LCCV). Instead of creating train-test splits with a large portion of training data, LCCV iteratively increases the number of training examples used for training. In the context of model selection, it eliminates models that can be safely dismissed from the candidate pool. We run a large scale experiment on the 67 datasets from the AutoML benchmark, and empirically show that LCCV in over 90% of the cases results in similar performance (at most 0.5% difference) as 10-fold CV, but provides additional insights on the behaviour of a given model. On top of this, LCCV achieves runtime reductions between 20% and over 50% on half of the 67 datasets from the AutoML benchmark. This can be incorporated in various AutoML frameworks, to speed up the internal evaluation of candidate models. As such, these results can be used orthogonally to other advances in the field of AutoML.

## 1. Introduction

Model validation is an important aspect in several learning approaches in which different models compete against each other. Automated machine learning (AutoML) tools make excessive use of validation techniques to assess the performance of machine learning pipelines (Thornton et al., 2013; Feurer et al., 2015; Olson et al., 2016; Mohr et al., 2018). But also specific learners like decision tables (Kohavi, 1995) use cross-validation to assess the performance of a whole model or parts of it. In any case, the underlying activity that employs validation is a *model selection* process. In the context of model selection, the cross-validation procedure can often be considered just an interchangeable function that provides some estimate for the quality of a model. That is, approaches adopting cross-validation such as decision tables or AutoML tools typically are agnostic to the particular choice of the cross-validation technique but are just interested in some kind of "robust" performance estimation. Typical choices to implement such a validation function are leave-one-out validation (LOO), k-fold cross-validation (kCV), and Monte-Carlo Cross-Validation (MCCV); we refer to Abu-Mostafa et al. (2012) for details.

This work is based on the observation that in situations with a high number of possibly costly evaluations, the above validation methods are often unnecessarily slow as they consider more data than necessary to estimate the generalization performance. While large training sets are *sometimes* necessary to estimate the generalization performance of a learner, in many cases the number of data points required to build the best possible model of a class is much lower. Then, large train folds unnecessarily slow down the validation process, sometimes substantially. For example, the error rate of a linear SVM on the NU-MERAI28.6 dataset is 48% when training with 4 000 instances but also when training with 90 000 instances. However, in the latter case, the evaluation is almost 50 times slower. We can economize runtime when the latter evaluation could be neglected.

In this paper, we propose an iterative cross-validation approach through the notion of learning curves. Learning curves express the prediction performance of the models produced by a learner for different numbers of examples available for learning. Validation is now done as follows. Instead of evaluating a learner just for one number of training examples (say 80% of the original dataset size), it is evaluated, in increasing order, at different so-called *anchor points* of training fold sizes (e.g., increasing size of 64, 128, 256, 512, ...). At each anchor point, several validations are conducted until a stability criterion is met. For each learner, we evaluate at each anchor point whether it is still feasible to improve over the so-far best-seen model. We do this based on the assumption that learning curves are convex. When extrapolating the learning curve in the most optimistic way does not yield an improvement over the best-seen model, the current model is dropped, economizing the runtime that was otherwise spent on larger anchor points. We dub this approach learning curve cross-validation (LCCV). While other authors have also worked with the concept of sub-sampling (Jamieson and Talwalkar, 2016; Li et al., 2017; Petrak, 2000; Provost et al., 1999; Sabharwal et al., 2016) or learning curves modelling (Baker et al., 2018; Domhan et al., 2015; Klein et al., 2017; Leite and Brazdil, 2010; van Rijn et al., 2015), these techniques typically have a higher emphasize on the runtime aspect. As such, these techniques have on one hand the possibility of disregarding a good model early, while on the other hand, they do have more time to evaluate other models. For a more extensive overview, we refer the reader to Appendix A. The main contribution of LCCV is that it employs a convexity assumption on learning curves, and using this convexity assumption more often selects the best model, at the cost of higher runtime compared to the aforementioned techniques.

## 2. Learning Curve Cross-Validation

The idea of learning curve cross-validation (LCCV) is to compute an empirical learning curve with performance observations only at some points. To gain insights about the true learning curve, we define a finite set of *anchor points* $S = (s_1, .., s_T)$ and validate learners using training data of the sizes in $S$. To obtain stable estimates for the performance at $s \in S$, several independent such validations will be conducted at each anchor point.

The LCCV algorithm is sketched in Alg. 1 (Appendix B) and works as follows. The algorithm iterates over *stages*, one for each anchor point in $S$ (in ascending order). In the stage for anchor point $s$ (lines 6-11), the learner is validated by drawing folds of training size $s$ and validate it against data not in those folds. These validations are repeated until a stopping criterion is met (cf. Sec. 2.1). Then it is checked whether the observations

are currently compatible with a convex learning curve model. If this is not the case, the algorithm steps back two stages and gathers more points in order to get a better picture and "repair" the currently non-convex view (l. 12). Otherwise, a decision is taken with respect to the three possible interpretations of the current learning curve. First, if it can be inferred that the performance at $s_T$ will not be competitive with $r$, LCCV returns $\perp$ (l. 14, cf. Sec. 2.2), indicating that this model will not be competitive compared to the best found model so far, neglecting evaluations on anchor points with more data. Second, if extrapolation of the learning curve suggests that the learner is competitive, then LCCV directly jumps to full validation of the candidate (l. 16, cf. Sec. 2.3). In any other case, we just keep evaluating in the next stage (l. 18). Finally, the estimate for $s_T$ is returned together with the confidence intervals for all anchor points. Note that since most learners are not incremental, each stage implies training from scratch. In spite of this issue, LCCV uses at with the set of exponentially increasing anchor points at most twice as much as the one of 10CV in the worst case, and the experiments show that it is often substantially faster than 10CV.

## 2.1 Repeated Evaluations for Convex Learning Curves

To decide whether or not the samples at an anchor point are sufficient, LCCV makes use of the *confidence bounds* of the sample mean at the anchor points. To compute such confidence bounds (l. 8 in the algorithm), we assume that observations at each anchor follow a normal distribution. Since the true standard deviation is not available for the estimate, we use the empirical one. We can then use the confidence interval to decide on the necessity to acquire more samples or stop the sampling process. After a maximum number of samples, we continue to the next anchor point regardless of the confidence interval. Once the width of the interval drops below some predefined threshold $\varepsilon$, we consider the estimate (sample mean) to be sufficiently accurate (l. 6). Since error rates reside in the unit interval in which we consider results basically identical if they differ on an order of less than $10^{-4}$, this value could be a choice for $\varepsilon$. While this is indeed an arguably good choice in the last stage, the "inner" stages do not require such high degree of certainty. To achieve this objective, a lose confidence interval is perfectly fine. Eventually, the required certainty in the inner stages is just a parameter, and we found it to work well with a generous value of 0.1.

## 2.2 Aborting a Validation

Based on the observations made so far, we might ask if it is still conceivable that the currently validated learner will have a performance of at least $r$ at the full dataset size. Once we are sure that we cannot reach that threshold $r$ anymore, we can return $\perp$ (l. 14).

The assumption that learning curves have a *convex* shape is key in deciding whether to abort the validation early. An important property following from convexity is that the *slope* of the learning curve is an *increasing* function (for measures that we want to minimize) that, for increasing training data sizes, approximates 0 from below (and sometimes even exceeds it). The crucial consequence is that the slope of the learning curve at some point $x$ is lower or equal than the slope at some "later" point $x' > x$. Formally, denote the learning curve as a function $f$. We have that $f'(x) < f'(x')$ if $x' > x$. Of course, as a learning curve is a sequence, it is not derivable in the strict sense, so we rather sloppily refer to its slope $f'$ as

the slope of the line that connects two neighbored values. In the empirical learning curve with only some anchor estimates, this means the following. If the sample means $v_i$ of the observations $O_i$ for a number $s_i \in S$ of training samples were perfect, then the slope of the learning curve after the last (currently observed) anchor point would be at least as high as the maximal slope observed before, which is just the slope of the last leg. That is,

$$f'(x) \geq \max_i \left\{ \frac{v_{i+1} - v_i}{s_{i+1} - s_i} \right\} = \frac{v_t - v_{t-1}}{s_t - s_{t-1}} \quad \forall x \geq s_t,$$

where $t$ is the number of completed stages. Perfect here means that the $v_i$ essentially *are* the true learning curve values at those anchor points.

However, the empirical estimates are imprecise, which might yield too pessimistic slopes and cause unjustified pruning. It is hence more sensible to rely on the most optimistic slopes possible under the determined confidence bounds as described in Sec. 2.1. To this end, let $C_i$ be the confidence intervals of the $v_i$ at the already examined anchors. Extrapolating from the *last* anchor point $s_t$, the best possible performance of this particular learner is

$$f(x') \geq f(s_t) - (|x'| - s_t) \left( \frac{\sup C_{t-1} - \inf C_t}{s_{t-1} - s_t} \right), \tag{1}$$

which can now be compared against $r$ to decide whether or not the execution should be stopped by choosing $|x'|$ to be the "maximum" training set size, e.g. 90% of the original dataset size if using 10CV as a reference.

## 2.3 Skipping Intermediate Evaluations

Unless we insist on evaluating at all anchor points for insight maximization, we should evaluate a learner on the full data once it becomes evident that it is definitely competitive. This is obviously the case if we have a *stable* (small confidence interval) anchor point score that is at least as good as $r$; in particular, it is automatically true for the first candidate. Of course, this only applies to non-incremental learners; for incremental learners, we just keep training to the end or stop at some point if the condition of Sec. 2.2 applies. Note that, in contrast to cancellation, we cannot lose a relevant candidate by jumping to the full evaluation. We might waste some computational time, but, unless a timeout applies, we cannot lose relevant candidates. Hence, we do not need strict guarantees for this decision as for cancellation. With this observation in mind, it indeed seems reasonable to *estimate* the performance of the learner on the full data and to jump to the full dataset size if this estimate is at least as good as $r$. A dozen of function classes have been proposed to model the behavior of learning curves (see, e.g., Domhan et al. (2015)), but one of the most established ones is the inverse power law (IPL) (Bard, 1974). The inverse power law allows us to model the learning curve as a function

$$\hat{f}(x) = a - bx^{-c}, \tag{2}$$

where $a, b, c > 0$ are positive real parameters. Given observations for at least three anchor points, we can fit a non-linear regression model using, for example, the Levenberg-Marquardt algorithm (Bard, 1974). After each stage, we can fit the parameters of the above model and check whether $\hat{f}(x') \leq r$, where $x'$ is again our reference training set size, e.g. 90% of the full data (l. 16 in the algorithm).

## 3. Evaluation

In this evaluation, we compare the effect of using LCCV instead of 10CV as validation methods inside a simple AutoML tool based on random search. To enable full reproducibility, the implementations of all experiments conducted here alongside the code to create the result figures and tables are available for the public[1].

Our evaluation measures the runtime of a random search AutoML tool that evaluates 1 000 candidate classifiers from the scikit-learn library (Pedregosa et al., 2011) using the two validation techniques LCCV and 10CV. The behavior of the random search is simulated by creating a randomly ordered sequence $A$ of 1 000 learners. The model selection technique (random AutoML tool) simply iterates over this sequence, calls the validation method for each candidate, and updates the currently best candidate based on the result. For LCCV, the currently best observation is passed as parameter $r$, and a candidate is discarded if $\perp$ is returned. For the sake of this evaluation, we simply used copies of 17 classifiers under default parametrization. Appendix C lists the used classifiers.

The performance of a validation method on a given dataset is the *average total runtime* of the random search using this validation method. Of course, the concrete runtime can depend on the concrete set of candidates but also, in the case of LCCV, their order. Hence, over 10 seeds, we generate different classifier *portfolios* to be validated by the techniques, measure their overall runtime (of course using the same classifier portfolio and identical order for both techniques), and form the average over them. As common in AutoML, we configured a timeout per validation. In these experiments, we set the timeout per validation to 60s. This timeout refers to the *full* validation of a learner and not, for instance, to the validation of a single fold. Put differently, 60s after the random search invoked the validation method, it interrupts validation and receives a partial result. For the case of 10CV, this result is the mean performance among the evaluated folds or `nan` if not even the first fold could be evaluated. For LCCV, the partial result is the average results of the biggest evaluated anchor point.

To obtain insights over different types of datasets, we ran the above experiments for all of the 67 datasets of the AutoML benchmark suite (Gijsbers et al., 2019). These datasets offer a broad spectrum of numbers of instances and attributes. All the datasets are published on the `openml.org` platform (Vanschoren et al., 2013). The reported "ground truth" performance of the chosen learner of each run is computed by an exhaustive MCCV. To this end, we form 100 bi-partitions of split size 90%/10% and use the 90% for training and 10% for testing in all of the 100 repetitions. The average error rate observed over these 100 partitions is used as the validation score. Note that we do not need anything like "test" data in this evaluation, because we are only interested in whether the LCCV can reproduce the same model selection as 10CV.

The results are summarized in Fig. 1. The boxplots capture, in this order, the average runtimes of LCCV and 10CV (leftmost figure), the absolute and relative *reductions* of runtimes in minutes when using LCCV compared to 10CV (central plots), and the deviations in the error rate of the eventually chosen model when using LCCV compared to 10CV (rightmost figure). More detailed insights and visualizations can be found in Appendix D. The first observation is dedicated to the last plot and confirms that, to a large degree,

---

1. `https://github.com/fmohr/lccv`

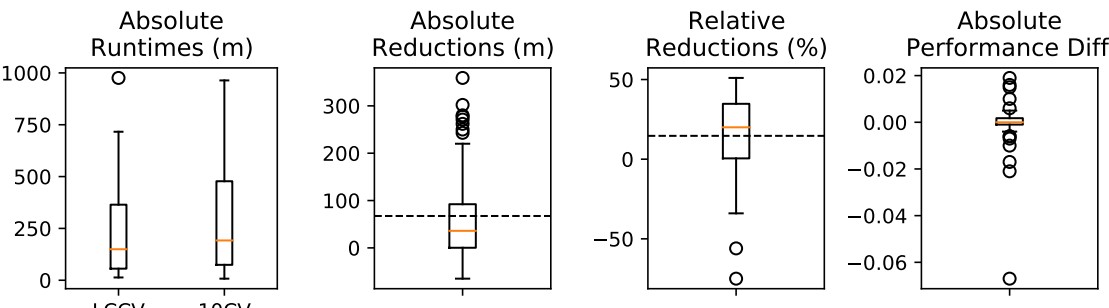

Figure 1: Comparison between LCCV and 10CV as validators in a random search.

LCCV and 10CV produce solutions of similar quality. In all but two of the 67 cases, the performance difference between LCCV and 10CV is within a range of less than 0.01, and among these the number of times when LCCV is even better than 10CV is balanced. The two cases in which LCCV failed to reproduce the result of 10CV are due to a slightly non-convex behavior of the optimal learner, which we discuss below in more detail. In this light, the runtime reductions achieved by LCCV over 10CV appear to be quite decent. Looking at the absolute reductions (left center plot), we see that the mean reduction (dashed line) is around 70 minutes. More precisely, this is a reduction from 290m runtime for 10CV compared to an average runtime of 222m for LCCV, which corresponds to a relative average runtime reduction of 15%. Even the absolute median runtime reduction is 36m, so on half of the datasets the overall runtime was reduced by at least 36 minutes and up to over 5 *hours*, which are relative reductions between 20% and 50%.

## 4. Conclusion

We presented LCCV, a technique to validate learning algorithms based on learning curves. In contrast to other evaluation methods that leverage learning curves, LCCV is designed to only prune once it can clearly no longer improve upon the current best candidate. Based on a convexity assumption, which turns out to hold almost always in practice, candidates are pruned once they can provably no longer improve upon the current best candidate. This makes LCCV potentially slower, but more reliable than other learning curve-based approaches, whereas it is faster and equally reliable as vanilla cross-validation methods.

We ran preliminary experiments that showed that LCCV outperforms 10CV in many cases in terms of runtime of a random-based model selection algorithm that employs these methods for validation. Reductions are on the order of up to between 20% and 50%, which corresponds to 60 minutes reduction on average and up to over 5 hours in some cases. We emphasize that LCCV is a contribution that is complementary to other efforts in AutoML and can be used with many of the tools in this field.

Future work will focus on integrating LCCV into various AutoML workbenches, such as Auto-sklearn and ML-Plan. Indeed, having enabling these toolboxes to speed up the individual evaluations, has the potential to further push the state-of-the-art of AutoML research.

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

## Appendix A. Related Work

Model selection is at the heart of many data science approaches. When provided with a new dataset, a data scientist is confronted with the question of which model to apply to this. This problem is typically abstracted as *algorithm selection* or *hyperparameter optimization problem*. To properly deploy model selection methods, there are three important components:

1. A *configuration space* which specifies which set of algorithms is complementary and should be considered

2. A *search procedure* which determines the order in which these algorithms are considered

3. An *evaluation mechanism* which assesses the quality of a certain algorithm

Most research addresses the question how to efficiently search the configuration space, leading to a wide range of methods such as random search (Bergstra and Bengio, 2012), Bayesian Optimization (Bergstra et al., 2011; Hutter et al., 2011; Snoek et al., 2012), evolutionary optimization (Loshchilov and Hutter, 2016), meta-learning (Brazdil et al., 2008; Pinto et al., 2016) and planning-based methods (Mohr et al., 2018).

Our work aims to make the evaluation mechanism faster, while at the same time avoid compromising the performance of the algorithm selection procedure. As such, it can be applied orthogonal to the many advances made on the components of the configuration space and the search procedure.

The typical methods used as evaluation mechanisms are using classical methods such as a holdout set, cross-validation, leave-one-out cross-validation, and bootstrapping. This can be sped up by applying racing methods, i.e., to stop evaluation of a given model once a statistical test excludes the possibility of improving over the best seen so far. Some notable methods that make use of this are ROAR (Hutter et al., 2011) and iRace (López-Ibáñez et al., 2016). Feurer and Hutter (2018) focus on setting the hyperparameters of AutoML methods, among others selecting the evaluation mechanism. Additionally, model selection methods are accelerated by considering only subsets of the data. By sampling various subsets of the dataset of increasing size, one can construct a learning curve. While these methods at their core have remained largely unchanged, there are two directions of research building upon this basis: (i) model-free learning curves, and (ii) learning curve prediction models.

**Model-free learning curves:** The simplest thing one could imagine is training and evaluating a model based upon a small fraction of the data (Petrak, 2000). Provost et al. (1999) propose progressive sampling methods using batches of increasing sample sizes (which we also leverage in our work) and propose mechanisms for detecting whether a given algorithm has already converged. However, the proposed convergence detection mechanism does not take into account randomness from selecting a given set of sub-samples, making the method fast but at risk of terminating early. More recently, successive halving addresses the model selection problem as a bandit-based problem, that can be solved by progressive sampling (Jamieson and Talwalkar, 2016). All models are evaluated on a small number of instances, and the best models get a progressively increasing number of instances. While

this method yields good performance, it does not take into account the development of learning curves, e.g., some learners might be slow starters, and will only perform well when supplied with a large number of instances. For example, the extra tree classifier (Geurts et al., 2006) is the best algorithm on the PHISHINGWEBSITES dataset when training with all (10 000) instances but the *worst* when using 1 000 or fewer instances; it will be discarded by successive halving. Hyperband aims to address this by building a loop around successive halving, allowing learners to start at various budgets (Li et al., 2017). Progressive sampling methods can also be integrated with AutoML systems, such as TPOT Gijsbers et al. (2018). Sabharwal et al. (2016) propose a novel method called Data Allocation using Upper Bounds, which aims to select a classifier that obtains near-optimal accuracy when trained on the full dataset while minimizing the cost of misallocated samples. While the aforementioned methods all work well in practice, these are all greedy in the fact that they might disregard a certain algorithm too fast, leading to fast model selection but sometimes sub-optimal performances.

Another important aspect is that the knowledge of the whole portfolio plays a key role in successive halving and Hyperband. Other than our approach, which is a *validation* algorithm and does not have knowledge about the portfolio to be evaluated (and makes no global decisions on budgets), successive halving assume that the portfolio is already defined and given, and Hyperband provides such portfolio definitions. In contrast, our approach will just receive a sequence of candidates for validation, and this gives more flexibility to the approaches that want to use it.

**Learning curve prediction models:** A model can be trained based on the learning curve, predicting how it will evolve. Domhan et al. (2015) propose a set of parametric formulas to which can be fitted so that they model the learning curve of a learner. They employ the method specifically to neural networks, and the learning curve is constructed based on epochs, rather than instances. This allows for more information about earlier stages of the learning curve without the need to invest additional runtime. By selecting the best fitting parametric model, they can predict the performance of a certain model for a hypothetical increased number of instances. Klein et al. (2017) build upon this work by incorporating the concept of Bayesian neural networks.

When having a set of historic learning curves at disposition, one can efficiently relate a given learning curve to an earlier seen learning curve, and use that to make predictions about the learning curve at hand. Leite and Brazdil (2010) employ a k-NN-based model based on learning curves for a certain dataset to determine which datasets are similar to the dataset at hand. This approach was later extended by van Rijn et al. (2015), to also select algorithms fast.

Most similar to our approach, Baker et al. (2018) proposed a model-based version of Hyperband. They train a model to predict, based on the performance of the last sample, whether the current model can still improve upon the best-seen model so far. Like all model-based learning curve methods, this requires vast amounts of meta-data, to train the meta-model. And like the aforementioned model-free approaches, these model-based approaches are at risk of terminating a good algorithm too early. In contrast to these methods, our method is model-free and always selects the optimal algorithm based on a small set of reasonable assumptions.

## Appendix B. Pseudo Code

---

**Algorithm 1:** LCCV: LearningCurveCrossValidation

---

**1** $(s_1, .., s_T) \leftarrow$ initialize anchor points according to min_exp and data size;
**2** $(C_1, .., C_T) \leftarrow$ initialize confidence intervals as $[0, 1]$ each;
**3** $t \leftarrow 1$;
**4** **while** $t \leq T \wedge (\sup C_T - \inf C_T > \varepsilon) \wedge |O_T| < n$ **do**
**5**     repair_convexity $\leftarrow$ false;

     /* work in stage $t$: gather samples at current anchor point $s_t$    */
**6**     **while** $\sup C_t - \inf C_t > \varepsilon \wedge |O_t| < n \wedge \neg$ *repair_convexity* **do**
**7**        add sample for $s_t$ training points to $O_t$;
**8**        update confidence interval $C_t$;
**9**        **if** $t > 1$ **then**   $\sigma_{t-1} = (\sup C_{t-1} - \inf C_t)/(s_{t-1} - s_t)$ ;
**10**       **if** $t > 2 \wedge \sigma_{t-1} < \sigma_{t-2} \wedge |O_{t-1}| < n$ **then**
**11**          repair_convexity $\leftarrow$ true;

     /* Decide how to proceed from this anchor point            */
**12**     **if** *repair_convexity* **then**
**13**       $t \leftarrow t - 1$;
**14**     **else if** *projected bound for $s_T$ is* $> r + \delta$ **then**
**15**       **return** $\bot$
**16**     **else if** $r = 1 \vee (t \geq 3 \ \wedge \text{IPL\_ESTIMATE}(s_T) \leq r)$ **then**
**17**       $t \leftarrow T$;
**18**     **else**
**19**       $t \leftarrow t + 1$

**20** **return** $\langle mean(C_T), (C_1, .., C_T) \rangle$

---

## Appendix C. Classifiers used in the Evaluation

The list of classifiers applied from the scikit-learn library Pedregosa et al. (2011) is as follows: LinearSVC, DecisionTreeClassifier, ExtraTreeClassifier, LogisticRegression, PassiveAggressiveClassifier, Perceptron, RidgeClassifier, SGDClassifier, MLPClassifier, LinearDiscriminantAnalysis, QuadraticDiscriminantAnalysis, BernoulliNB, MultinomialNB, KNeighborsClassifier, ExtraTreesClassifier, RandomForestClassifier, GradientBoostingClassifier

     All classifiers have been considered under default configuration exclusively. In the evaluations of Sec. 3 with portfolios of 1000 learners, each learner in the portfolio corresponds to a copy of one of the above. Note that since we just use these learners as representatives of some learners produced by the AutoML algorithm, it is unnecessary to consider truly different learner parametrizations. One can just consider these copies to be "other" algorithms that "incidently" have the same (or very similar) error rates and runtimes as others in the portfolio.

# Appendix D. Detailed Results of the Random Search

The detailed results of the two validation methods per dataset are shown in Fig. 2. The semantics of this figure is as follows. Each pair of points connected by a dashed line stand for evaluations of a dataset. In this pair, the blue point shows the mean runtime of the random search applying LCCV (x-axis) and the eventual mean error rate (y-axis). The orange point shows the same information when validating with 10CV. Note that the runtimes are shown on a log-scale. The dashed lines connecting a pair of points are green if LCCV is faster than CV and red otherwise. The line is labeled with the dataset id and the absolute (and relative) difference in runtime. The *vertical* dashed lines are visual aids in the log-scale to mark runtimes of 30 minutes, 1 hour, and 10 hours.

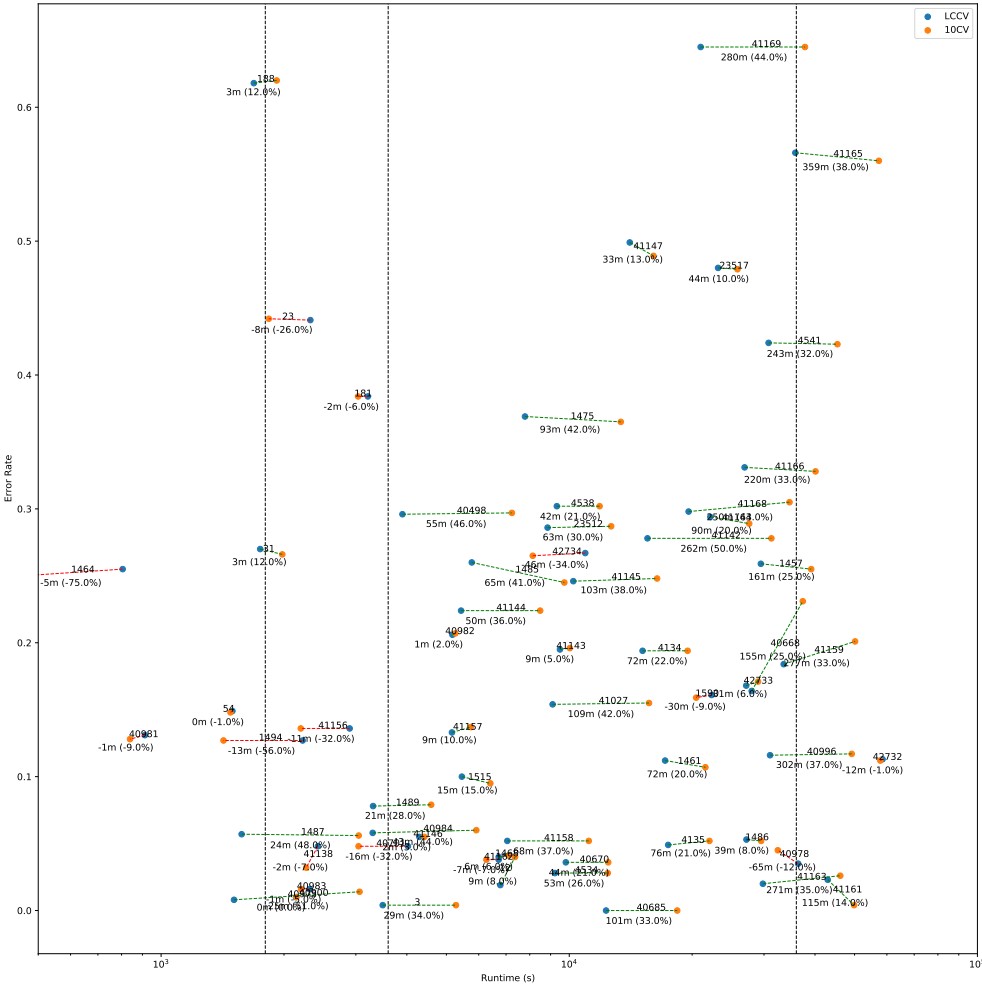

Figure 2: Visual comparison of LCCV (blue) and random search (orange). The x-axis displays the runtime in seconds, the y-axes displays the error rate. The dashed line indicates which LCCV results and random search results are performed on the same dataset.

