# OpenReview forum: "Towards Model Selection using Learning Curve Cross-Validation"
_ICML.cc/2021/Workshop/AutoML — AutoML@ICML2021 Poster_

### Official Review · Reviewer_evdJ · 2021-06-13
**Insufficient novelty and presentation needs to be improved**

**Rating:** 4
**Confidence:** 4

**Review:**

The paper proposes an alternative model selection mechanism in a data efficient way.

Overall, the paper is poorly written. The main idea should be better presented clearly by having a figure that describes it, as well as examples to show insights on why it makes sense.

The method is based on various assumptions, such as the inverse power law fitting, without sufficient justification. Why should such a learning curve follow power law? Theoretical or experimental basis for this is required.

Runtime reduction is not the most meaningful metric as it depends on the hardware, and parallelization. I suggest focusing on a better metric that is agnostic to these.

Why is the comparison CV fold 10? How about no cross validation, and lower (3 or 5 fold) benchmarks?

---

### Official Review · Reviewer_5R1b · 2021-06-15
**Reasonable model training time saving scheme, but lacks some clarity and justification**

**Rating:** 5
**Confidence:** 4

**Review:**

At a high level, LCCV is trying to decide whether to increase the size of the training data (and thus allocate more compute resources) for a given model (and in general a model configuration). It uses a statistical approach (a Gaussian assumption) and a convexity assumption to measure confidence of projected model performance.

A sketch of LCCV:
1. Split the data into (exponentially) increasing subsets
2. for each subset:
    a. start/resume training the model
    b. evaluate the model on unseen data
    c. measure some confidence of validation performace
    d. determine whether the loss function is convex or not (keep training/validating if not)
    e. determine if the loss has flattened enough to stop
    f. estimate model performance on all the training data (using an IPL model)

The results show that final model performance is not significantly different (a good thing), but model training time is reduced by 15% on average and 36 to 300 minutes on half of the datasets.

In the Conclusion, the authors claim that LCCV is possibly slower, but more reliable than other methods that are not explored empirically or theoretically in the body of the paper. Either compare and contrast other approaches in the main body, or leave it out of the conclusion, preferably the former (which is understandably more work).

The main thesis has two parts:
1. Use small, but then increasing subsets of data to train models so that early termination can happen even sooner
2. A probabilistic loss estimation scheme to decide whether to terminate model training early

The convexity assumption is very reasonable.

The explanation of the algorithm and the formal algorithm outline are not written clearly enough to fully understand the authors' proposed method.

Because the contribution has two parts, it is unclear which part is causing the reduction in training time. Layered TPOT [1] also proposes using increasing subsets of data to terminate early. The probabilistic/IPL approach seems reasonable, but lacks more rigorous justification/proof/analysis. There may exist more robust/well-understood probabilistic time-series forecasting approaches that could be used.

Overall, the approach seems based in reasonable decisions and assumptions. However, the explanation is not precise enough and the justification needs improvement.

References:
[1] Gijsbers, Pieter, Joaquin Vanschoren, and Randal S. Olson. "Layered TPOT: Speeding up tree-based pipeline optimization." arXiv preprint arXiv:1801.06007 (2018).

Notes taken while reading the paper:

(The author names appear on the pdf, so this is not a doubly blind review.)

The claim that large amounts of data is not necessary to validate a model should be justed theoretically, empirically, or with related work. Where did the SVM/Numerai28.6 example numbers come from? This particular example gives one data point supporting the claim, but how do we know this is true in general or at least more often than not?

Many learning curves express generalization performance for increasing repetitions of the same number of training instances, not different numbers of training examples.

Convexity of learning curves is a reasonable assumption based on this reviewer's experience, but perhaps some justification would be good.

Algorithm 1 is not defined precisely enough. What is min_exp? How are (s_1, ..., s_T) initialized? What does "update confidence interval C_t" mean? When presenting a new algorithm, a precise psuedo-code sketch is invaluable.

What is the parameter r?

Using x' in conjunction with f' is a little confusing, maybe consider just x and y with y > x.

---

### Official Review · Reviewer_oAdT · 2021-06-17
**A very well-written paper with an important discussion**

**Rating:** 7
**Confidence:** 3

**Review:**

This paper addresses sampling techniques and how incremental sampling can be more efficient than traditional approaches. The topic is relevant, the paper is well-written, and results are compelling. Though the investigation is proof-of-concept, I believe it suits the workshop and will stir very interesting discussions. Importantly, code is available for reproduction.

A few, minor comments:
- The paper is well-written, but still has typos and grammar errors. Please review it thoroughly.
- I missed how the authors would use the learning curve to obtain further insights on the validation process, as argued in the introduction. This should be clarified.
- Appendix A list algorithm selection and hyperparameter optimization as interchangeable terms. This is not correct.
- It would be interesting to contrast the scenario for (non)incremental learners. How would an incremental learner benefit differently from the increasing size anchor points?
- Appendix D brings a lot of information, but the discussion is very brief. Please improve it.

---

### Decision · Program_Chairs · 2021-06-21

**Decision:**

Accept (Poster)

**Comment:**

The authors present a novel method for model selection that is a replacement for k-fold cross-validation. This new method is interesting and relevant to the workshop.

The main concerns from the reviewers is that the method should be presented more clearly, and the assumptions should be verified more explicitly. In particular, Algorithm 1 should be more precise, the main text can be improved, and the assumptions should be justified experimentally.

We recommend acceptance based on the novelty of the method and the relevance to the workshop. For the camera ready version, we encourage the authors to address the concerns mentioned by the reviewers.